# Individual, health facility and wider health system factors contributing to maternal deaths in Africa: A scoping review

**Francis G. Muriithi**[1]*, **Aduragbemi Banke-Thomas**[2], **Ruth Gakuo**[3], **Kia Pope**[4], **Arri Coomarasamy**[1], **Ioannis D. Gallos**[1]

1 WHO Collaborating Centre for Global Women's Health, Institute of Metabolism and Systems Research, University of Birmingham, Edgbaston, Birmingham, United Kingdom, 2 School of Human Sciences, University of Greenwich, Old Royal Naval College, Park Row, Greenwich, London, United Kingdom, 3 School of Nursing, University of Derby, Derby, United Kingdom, 4 Nottingham Medical School, Queen's Medical Centre, Nottingham, United Kingdom

* fgm911@student.bham.ac.uk

**Data Availability Statement:** All the data relating to this manuscript are fully available and without restriction as supplementary files.

## Abstract

The number of women dying during pregnancy and after childbirth remains unacceptably high, with African countries showing the slowest decline. The leading causes of maternal deaths in Africa are preventable direct obstetric causes such as haemorrhage, infection, hypertension, unsafe abortion, and obstructed labour. There is an information gap on factors contributing to maternal deaths in Africa. Our objective was to identify these contributing factors and assess the frequency of their reporting in published literature. We followed the Arksey and O'Malley methodological framework for scoping reviews. We searched six electronic bibliographic databases: MEDLINE, SCOPUS, African Index Medicus, African Journals Online (AJOL), French humanities and social sciences databases, and Web of Science. We included articles published between 1987 and 2021 without language restriction. Our conceptual framework was informed by a combination of the socio-ecological model, the three delays conceptual framework for analysing the determinants of maternal mortality and the signal functions of emergency obstetric care. We included 104 articles from 27 African countries. The most frequently reported contributory factors by level were: (1) Individual—level: Delay in deciding to seek help and in recognition of danger signs (37.5% of articles), (2) Health facility—level: Suboptimal service delivery relating to triage, monitoring, and referral (80.8% of articles) and (3) Wider health system—level: Transport to and between health facilities (84.6% of articles). Our findings indicate that health facility—level factors were the most frequently reported contributing factors to maternal deaths in Africa. There is a lack of data from some African countries, especially those countries with armed conflict currently or in the recent past. Information gaps exist in the following areas: Statistical significance of each contributing factor and whether contributing factors alone adequately explain the variations in maternal mortality ratios (MMR) seen between countries and at sub-national levels.

**Funding:** This study was supported by the Institute of Global Innovation (IGI), University of Birmingham, as part of Dr Francis G Muriithi's Doctoral Research Fellowship. AC holds a Bill and Melinda Gates Foundation Grant. Award number INV-001393. The Institute of Global Innovation and Bill and Melinda Gates Foundation had no role in the study design, data collection and analysis, interpretation of findings, manuscript preparation, or the decision to publish.

**Competing interests:** The authors have declared that no competing interests exist.

## Introduction

The number of women dying during and after pregnancy and childbirth remains unacceptably high, especially in African countries [1,2]. As of 2017, global estimates reported that 830 women die daily of pregnancy-related conditions, with Africa contributing two-thirds of these deaths [2]. When pregnant women die, there are negative consequences on their surviving neonates, who become orphaned children, their families, communities, and societies, often lasting beyond one generation [3,4]. Tackling the causes and factors contributing to Africa's preventable maternal mortality burden may contribute positively to human development [5].

The leading causes of maternal deaths in Africa are due to preventable direct obstetric causes, including obstetric haemorrhage, sepsis, hypertension, unsafe abortion, and obstructed labour [6–8]. Direct obstetric deaths result from obstetric complications of the pregnancy state (pregnancy, labour, and the puerperium), complications of medical or surgical interventions, omission of critical treatment or incorrect treatment, or a chain of events resulting from any of the above [9]. While the causes of maternal death remain largely similar between African countries, there is variation in the Maternal Mortality Ratio (MMR) between and within African countries [10]. Understanding the factors contributing to the variation in MMR at national and sub-national levels remains either understudied or underreported [11].

Since 1987, various global initiatives have collectively contributed to the reduction of preventable maternal deaths agenda worldwide. These initiatives include the Safe Motherhood Initiative (SMI), launched following a conference in Nairobi in 1987 and revised following a meeting in Colombo in 1997 [12,13]. Subsequently, SMI was merged with the Child Survival and Newborn initiatives to form the Partnership for Maternal, Newborn and Child Health (PMNCH) in 2005 [14]. There was an overlap with the Millennium Development Goals (MDGs) initiative, which is credited with achieving a 45% reduction in maternal mortality in sub-Saharan Africa between 1990 and 2015 [2]. However, this success was not uniform, and many African countries failed to meet the set target of a 75% reduction in MMR [5]. Other initiatives include the publication of the first global strategy on maternal, newborn and child health (MNCH) in 2010 and Maternal Deaths Surveillance and Response (MDSR) in 2013 [15,16]. These initiatives have since been succeeded by the Sustainable Development Goals (SDG) initiative, whose target 3.1 aims to achieve a global average MMR of less than 70 per 100,000 live births by 2030 [17].

The lessons learnt from the previous initiatives, especially the MDGs, informed the formulation of the strategies toward ending preventable maternal mortality (EPMM), which aims to tackle maternal mortality by implementing key guiding principles such as empowerment of women, girls, families and communities, integration of maternal and newborn care, country ownership and a human rights perspective [18]. Promotion of good governance, less dependence on donor funding in favour of in-country funding, scaling up of evidence-based interventions and addressing the background and proximate factors that predispose women to maternal deaths were proposed as the key priorities for tackling maternal deaths in Africa [5].

So far, most articles that have explored the relationship between contributory factors and maternal deaths in Africa used statistically modelled population-level data from the WHO Global Health Observatory, Human Development Report, World Bank and United Nations bodies [11,19–24]. A single study utilised primary data from hospital-based studies [25]. Studies of maternal mortality that use data sources based on statistically modelled MMR estimates or population-level data are not always as accurate as estimates derived from primary studies which explore individual patient-level data [26]. Therefore, there is a rationale for this scoping review and for reviewing individual patient-level primary studies for factors contributing to

maternal deaths and comparing the findings with those based on statistically modelled population-level estimates.

An exploration of published literature may identify the most frequently reported contributing factors and identify potential gaps that may better inform policy formulation, strategic prioritisation of interventions, resource allocation, and future research [5,27]. This scoping review aims to identify the factors contributing to maternal deaths in Africa and assess the frequency of their reporting in published literature.

## Methods

### Overview

We conducted a systematic search to identify published literature on the factors contributing to maternal deaths in Africa. Our scoping review was done following the Arksey and O'Malley methodological framework [28]. We have reported our scoping review following the Preferred Reporting Items for Systematic Reviews and Meta-Analyses Extension for Scoping Reviews (PRISMA-ScR) guidelines [29]. The completed PRISMA-ScR checklist is appended as S1 Checklist.

### Search strategy and selection criteria

We systematically searched six electronic databases: PubMed, Scopus, African Index Medicus, African Journals Online (AJOL), French humanities and social sciences databases, and Web of Science. We used the search terms maternal death*, determinant*, factor*, Africa* (or the individual country name) and their synonyms. The search strategy and output are presented in S1 Table. All databases were searched on September 16, 2021, and we updated the search on October 8, 2021. We scrutinised the reference lists of published review articles for additional relevant publications of primary studies. We included publications from any African country without language restriction. Articles of any study design were included at the outset.

We exported all the articles into the EndNote version 20 (Clarivate Philadelphia, PA, USA) reference management software [30]. After collation, we uploaded them onto Covidence (Covidence, Melbourne, Australia), a systematic review management software [31]. We used EndNote to collate articles from the various sources and Covidence for deduplication, article screening and assessment for eligibility. Two reviewers pilot tested the screening criteria on a random sample of articles and abstracts to ensure consistency. Subsequently, two reviewers independently screened the articles in the title and abstract and full-text screening stages. We resolved any discrepancies by discussion between the two reviewers or by the adjudication of a third reviewer. We excluded inaccessible articles if no full-text versions were received from the corresponding authors or the University of Birmingham Library service.

We included articles reporting on primary studies and excluded case reports, commentaries, ecological studies, and systematic reviews (study design criterion). We examined the reference lists of any excluded systematic reviews for relevant studies. We restricted our review to articles published between 1987 and 2021 (time frame restriction criterion) to coincide with the period after the commencement of the Safe Motherhood Initiative [32,33]. We included articles on maternal deaths (population criterion), describing factors contributing to (determinants) maternal death (concept criterion) that occurred in African countries (context criterion). Duplicate articles were excluded. All included and excluded articles are presented in S1 and S2 Data, respectively.

### Data extraction, analysis, and synthesis

Two reviewers independently extracted data from included studies using a piloted extraction google form. Any discrepancies were resolved by discussion between the two reviewers or by the adjudication of a third reviewer. The data variables extracted from each study are presented in S1 Data. We adapted the WHO definition of determinants of health to define a determinant of maternal death as any personal, social, economic, or environmental factor influencing the occurrence of maternal death [34]. Any factor that was reported as contributory to maternal death was extracted. The conceptual framework we used to summarise our findings was informed by the socio-ecological model (SEM), the three delays framework for analysing the determinants of maternal mortality and the signal functions of emergency obstetric care [35–39]. The SEM is a theory-based approach to understanding how an individual interacts with other factors around them that influences their behaviour and choices [39]. The delays framework identifies three phases (delay in deciding to seek care, delay in reaching a health facility and delay in receiving adequate care once at the health facility) as important determinants of maternal health outcomes, including maternal deaths [35,36]. Signal functions for emergency obstetric care are a list of medical and surgical interventions that healthcare facilities are expected to provide so as to effectively prevent and treat direct obstetric complications [37,38]. Our conceptual framework is summarised in Fig 1.

Based on our conceptual framework, we generated a list of factors contributing to maternal deaths and synthesised them into three categories: Individual, health facility and wider health system levels. This categorisation method makes it easier to make strategic choices for intervention, especially in diverse country settings with scarce resources and a multiplicity of factors contributing to maternal deaths [40]. We adopted a descriptive analysis approach.

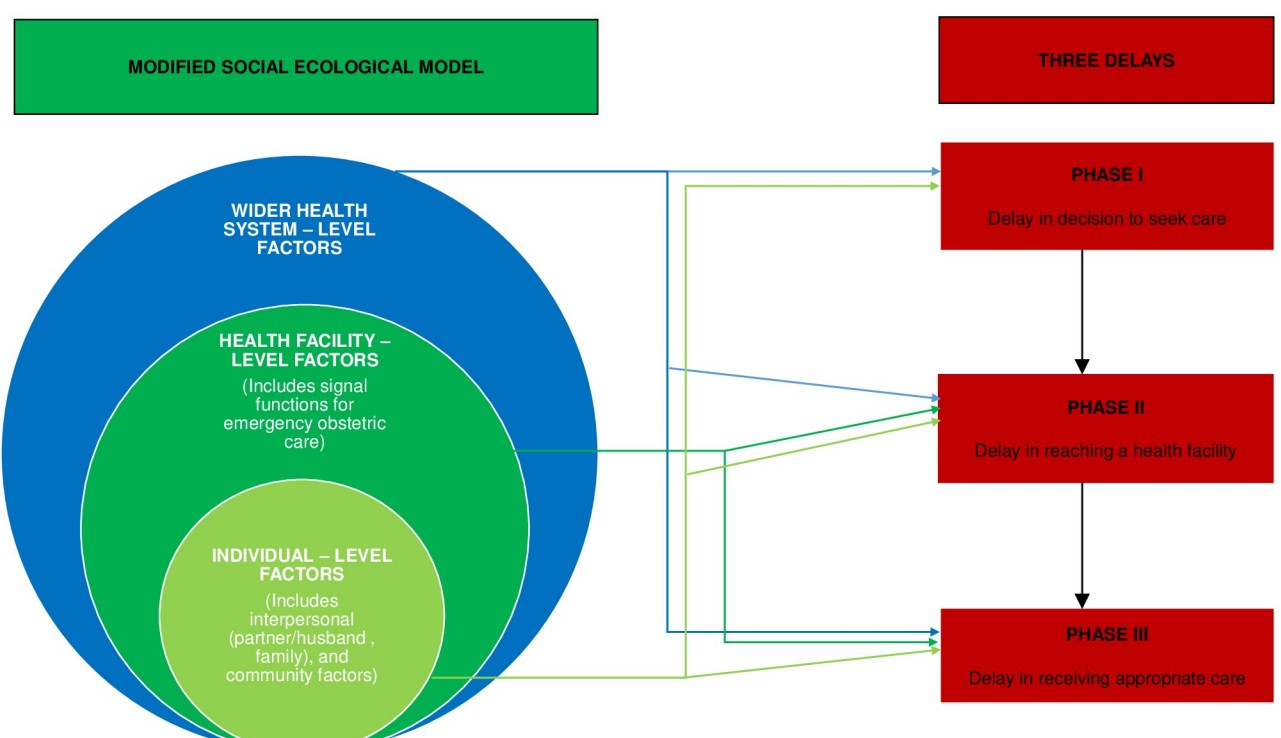

**Fig 1. Conceptual framework.** A modification of the social-ecological model into three levels of contributing factors: Individual, health facility and wider health system levels and their linkage to the three delays framework.

The protocol was developed before the review was done and is published on the Open Science Framework registries [41].

### Role of the funding source

This study was supported by the Institute of Global Innovation (IGI), University of Birmingham, as part of Dr Francis G Muriithi's Doctoral Research Fellowship. Professor Arri Coomarasamy holds a Bill and Melinda Gates Foundation Grant. Award number INV-001393. The Institute of Global Innovation and Bill and Melinda Gates Foundation had no role in the study design, data collection and analysis, interpretation of findings, manuscript preparation, or the decision to publish.

## Results

As outlined in the Prisma flow chart, our systematic search yielded 4436 articles, of which 3743 remained after duplicates were removed. A further 3327 articles were removed after the title and abstract screening process. The full texts of the remaining 416 articles were screened, the full texts of 30 articles were inaccessible, and a further 282 were ineligible for inclusion. One hundred and four (104) articles were eligible for inclusion in the final analysis [42–146]. The article identification and selection process are illustrated in the Prisma Flow Chart. See Fig 2.

### Characteristics of included studies

We included 104 studies from 27 African countries with a cumulative total of 18,440 maternal deaths. The sample size in individual studies ranged from 7 to 3025 maternal deaths. The largest proportion of included articles, 22 (21.2%), was from Nigeria. The included articles were of various study designs, with descriptive cross-sectional surveys and case series studies being the predominant study design at sixty (60.6%) per cent (63 of 104 articles). A review of medical records was the most common method used to investigate the contributing factors in 52.9% of included articles. The included articles represented maternal deaths across health facility and community levels, with the former being the most predominant at 77.9%. Most articles reported maternal deaths across the antenatal, intrapartum, and postpartum phases of pregnancy and puerperium. All the characteristics of included studies are presented as a S1 Data. A summary of the main characteristics of the included studies is presented in Table 1.

Table 1. A summary of the characteristics of the included studies by study design, the method used to investigate the maternal deaths, the location where the investigated maternal deaths occurred and the stage of pregnancy at the time of death. The n (%) represents the number of articles and the proportion as a percentage, with a denominator being the total number of included articles (N = 104). The last column consists of the reference articles for each characteristic.

The distribution of included articles by country was as follows:, Algeria [60], Benin [78], Burkina Faso [122,143], Cameroon [99,133], Democratic Republic of Congo [55,82], Egypt [105], Eritrea [102], Ethiopia [57,62,71,72,74,86,130,131], Ghana [49,52,53,58,69,141], Guinea-Bissau [76], Kenya [63,91,110,123,142,145], Madagascar [79], Malawi [80,83,100,126,140], Morocco [44], Mozambique [59], Nigeria [42,43,45,47,48,54,56,65,67,89,98,116–121,124,128,136, 138,146],Rwanda [132], Senegal [70], Sierra Leone [84], Somalia [46], South Africa [61,85, 90,93–95,101,103,106–108,134], Sudan [50,104], Tanzania [64,68,75,77,87,88,92,112,113, 125,129,137,144], Togo [135], Uganda [51,97,114,127,139], Zambia [73,109], Zimbabwe [66,96,111,115].

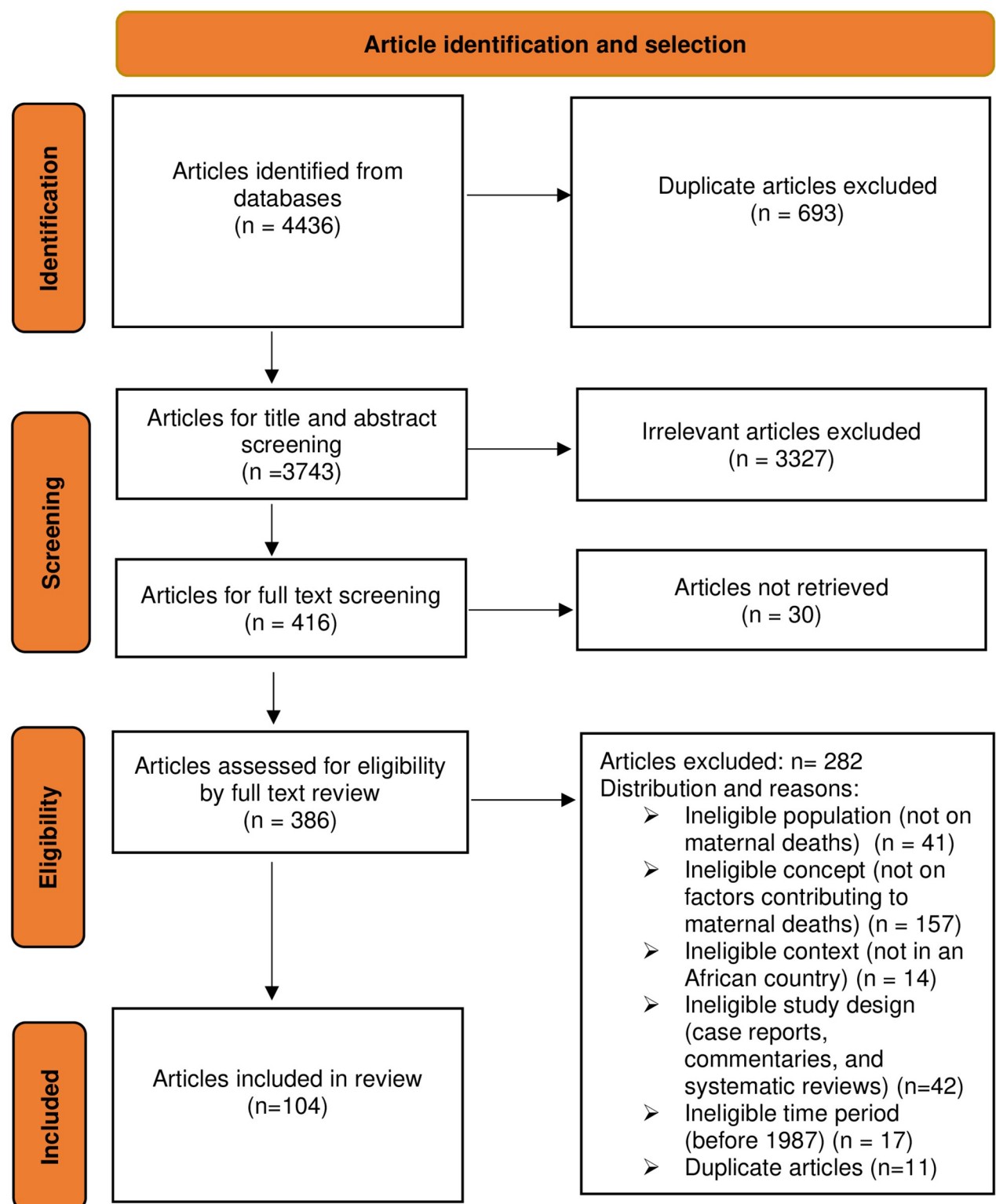

**Fig 2. Prisma flow chart of the scoping review process.** The flow chart illustrates the stages of article processing from identification from the various databases, screening using the inclusion and exclusion criteria, and inclusion in the review. In the final screening, primary articles reporting on maternal deaths (population), factors contributing to maternal deaths (concept) and within an African country (setting) were included.

**Table 1. Characteristics of included studies.**

| Characteristic | n (%) | References |
|---|---|---|
| **Study design** | | |
| Descriptive (Survey/Cross-sectional/Case-series) | 63 (60.6) | [43,44,47,48,54–58,61,65–67,71,72,74,76–78,82–85,89–91,94,97–99,101,106–108,110,112,113,115,117–120,123–138,140,141,143,144,146] |
| Analytic Observational (Cohort/Case-Control) | 16 (15.4) | [45,50,63,64,70,80,86,92,102,109,114,116,121,122,139,142] |
| Mixed methods | 10 (9.6) | [49,59,60,69,73,88,100,104,105,145] |
| Descriptive Qualitative | 9 (8.7) | [42,46,51,62,68,93,95,103,111] |
| Secondary data analysis | 3 (2.9) | [53,75,79] |
| Community-based case-referent design | 2 (1.9) | [96,137] |
| Confidential enquiry into maternal deaths | 1 (1.0) | [52] |
| **Grand Total** | **104 (100.0)** | |
| **Method used to investigate contributing factors** | | |
| Review of medical records | 55 (52.9) | [43,44,48,50,52,54–56,58,61,63,65,67,70,78,80,82,83,85,86,89–91,94,96,97,102,107,108,110,112–114,116–126,131,133,135,136,138–140,142–144,146] |
| Secondary data/database analysis | 14 (13.5) | [45,53,59,71,79,98,99,101,106,115,127,128,132,134] |
| Review of medical records and interviews (unspecified format) | 10 (9.6) | [49,64,66,68,69,87,100,109,137,141] |
| Review of medical records and structured interviews | 6 (5.8) | [46,57,76,105,130,145] |
| Secondary data/database analysis and interviews (unspecified format) | 5 (4.8) | [47,74,75,77,92] |
| Interviews (unspecified format) | 5 (4.8) | [72,81,88,93,111] |
| Semi-structured interviews | 3 (2.9) | [51,95,104] |
| Structured interviews | 2 (1.9) | [42,103] |
| Secondary data/database analysis and structured interviews | 1 (1.0) | [129] |
| Review of medical records and semi-structured interviews | 1 (1.0) | [73] |
| Review of medical records,structured and interviews (unspecified format) | 1 (1.0) | [84] |
| Secondary data/database analysis and semi-structured interviews | 1 (1.0) | [60] |
| **Grand Total** | **104 (100.0)** | |
| **Location of maternal deaths** | | |
| Facility | 81 (77.9) | [43–45,48,50–52,54–63,65–67,69–78,80,82,83,85–87,89–91,94,97,101,102,106–110,112–136,138–145] |
| Both facility and community | 11 (10.6) | [46,53,64,68,88,92,98–100,105,137] |
| Unspecified | 8 (7.7) | [47,49,84,93,95,103,111,146] |
| Community | 4 (3.8) | [42,79,96,104] |
| **Grand Total** | **104 (100.0)** | |
| **Stage of pregnancy at the time of death** | | |
| Antepartum, Intrapartum, Postpartum | 69 (66.3) | [46,50,51,54–68,71,72,75–78,82,83,85–90,92,97,100,101,105–110,116,119–144] |
| Unspecified | 26 (25.0) | [42,44,47,49,53,69,70,73,74,79,84,91,93,95,96,98,99,102–104,111–113,115,117,146] |
| Intrapartum and Postpartum only | 6 (5.8) | [45,52,94,114,118,145] |
| Intrapartum only | 1 (1.0) | [43] |
| Antenatal only | 1 (1.0) | [48] |
| Postpartum only | 1 (1.0) | [80] |
| **Grand Total** | **104 (100.0)** | |

No articles were eligible for inclusion from twenty-seven (27) countries: Angola, Botswana, Burundi, Cape Verde, Central African Republic, Chad, Comoros, Djibouti, Equatorial Guinea, Eswatini, Gabon, Guinea, Lesotho, Liberia, Libya, Mali, Mauritania, Mauritius, Namibia, Niger, Republic of the Congo, Republic of Côte d'Ivoire, Republic of the Gambia, Sao Tome and Principe, Seychelles, South Sudan and Tunisia.

## Individual-level factors contributing to maternal deaths

Ninety-six (95) articles reported individual-level contributing factors [42–48,50,52–58,60–62,64–70,72–80,82,83,86–134,136–140,142–146]. The factors we identified under this category are ranked and illustrated in Fig 3.

## Health facility-level factors contributing to maternal deaths

Seventy four (74) articles reported health facility-level contributing factors [42–44,46,47, 49,51,52,54–59,61–63,66,67,70,71,73,74,78,80,82,83,85,87,88,90,91,94,95,97,101,102,104–110,112–130,132,133,135,137,139–145].

The factors we identified under this category are ranked and illustrated in Fig 4.

They are ranked from the most frequently reported (triage, monitoring, and referral processes) to the least frequently reported (lack of intravenous fluids). The number and proportion n (%) of articles reporting a specific factor is on the X-axis. The total number of included

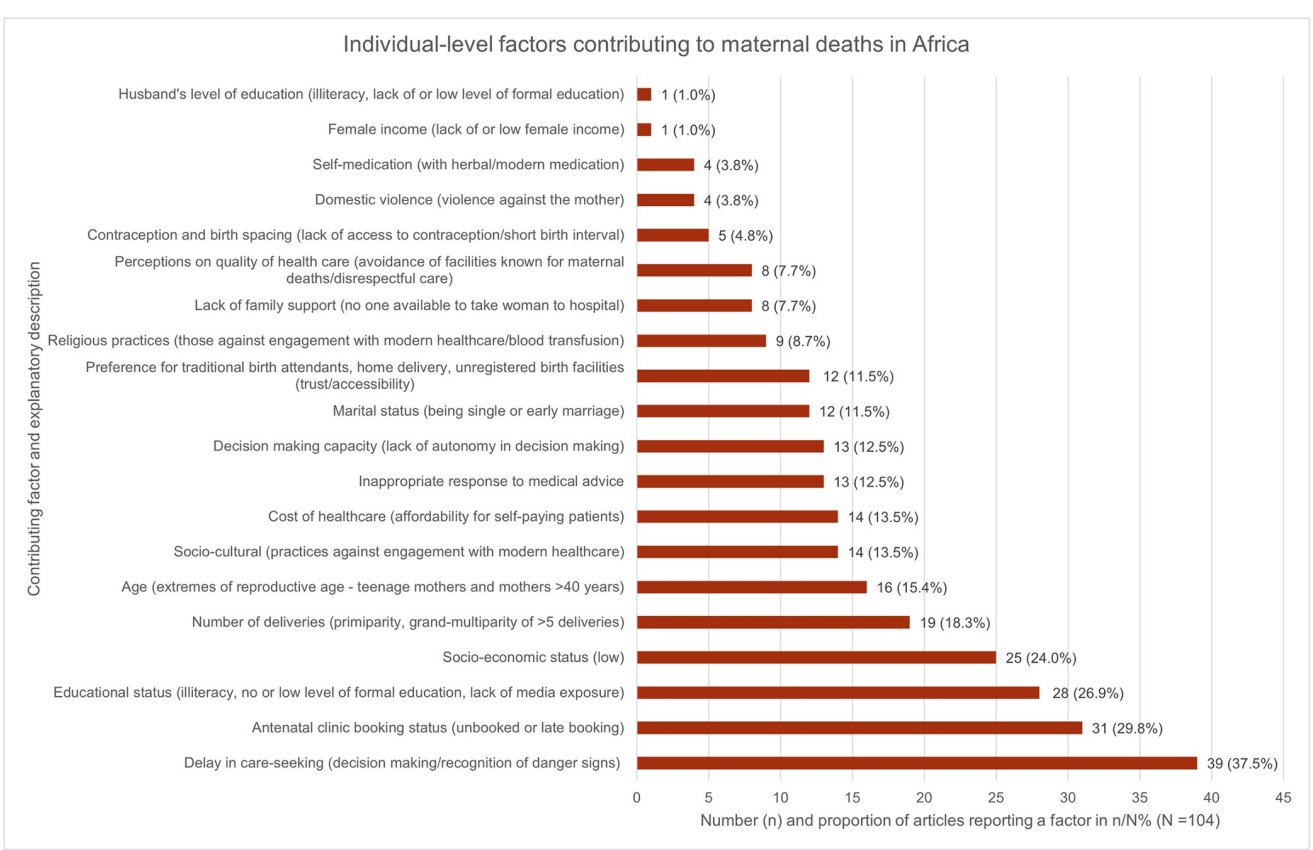

**Fig 3. Individual-level factors contributing to maternal deaths in Africa.** They are ranked from the most frequently reported (delay in care-seeking) to the least frequently reported (husband's level of education). The number and proportion n (%) of articles reporting a specific factor is on the X-axis. The total number of included articles (N = 104) is the denominator. The factors and their explanatory descriptions are on the Y-axis.

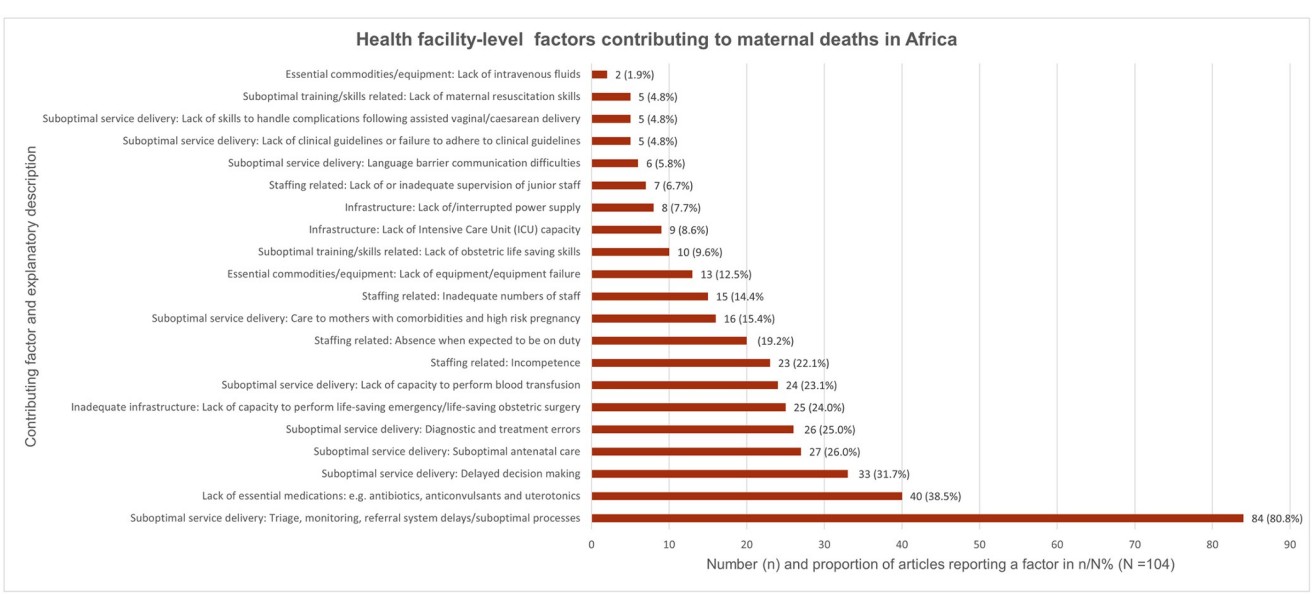

**Fig 4. Health-Facility level factors contributing to maternal deaths in Africa.**

articles (N = 104) is the denominator. The factors and their explanatory descriptions are on the Y-axis.

## Wider health system-level factors contributing to maternal deaths

Fifty-five (55) articles reported wider health system-level factors [42,44–47,52,56,57,59,62, 66,68,71–76,79,83–88,90,94,95,97–101,104,105,108–110,114,116,119,122,124,126–132, 137,138,140,144,145]. The factors we identified under this category are ranked in Fig 5.

## Comparison of the contributing factors

We compared the aggregated counts of all the contributing factors we identified across all the included papers. This analysis is presented in a table in S2 Table. Our main finding was that the health facility-level factors had the highest count of 403 contributory factors (49.0%), followed by individual-level factors with a count of 276 contributory factors (33.6%) and wider health system-level factors with a count of 143 contributory factors (17.4%). Under the health facility level, the most counts (226 contributing factors) were in the service delivery sub factor of triage, monitoring, and referral processes. The complete analysis is presented in a sunburst diagram illustrated in Fig 6.

## Discussion

### Main findings

This review examined articles focused on factors contributing to maternal deaths in 27 African countries. There were no eligible articles retrieved from 27 African countries.

The most frequently reported contributing factors were at the health facility level, with service delivery factors of triage, monitoring and referral being the most reported under this category. Delay in care seeking (decision-making and recognising danger signs) was the most frequently reported individual-level contributing factor. At the wider health system—level, transportation issues to and between health facilities were the most reported. We also found a

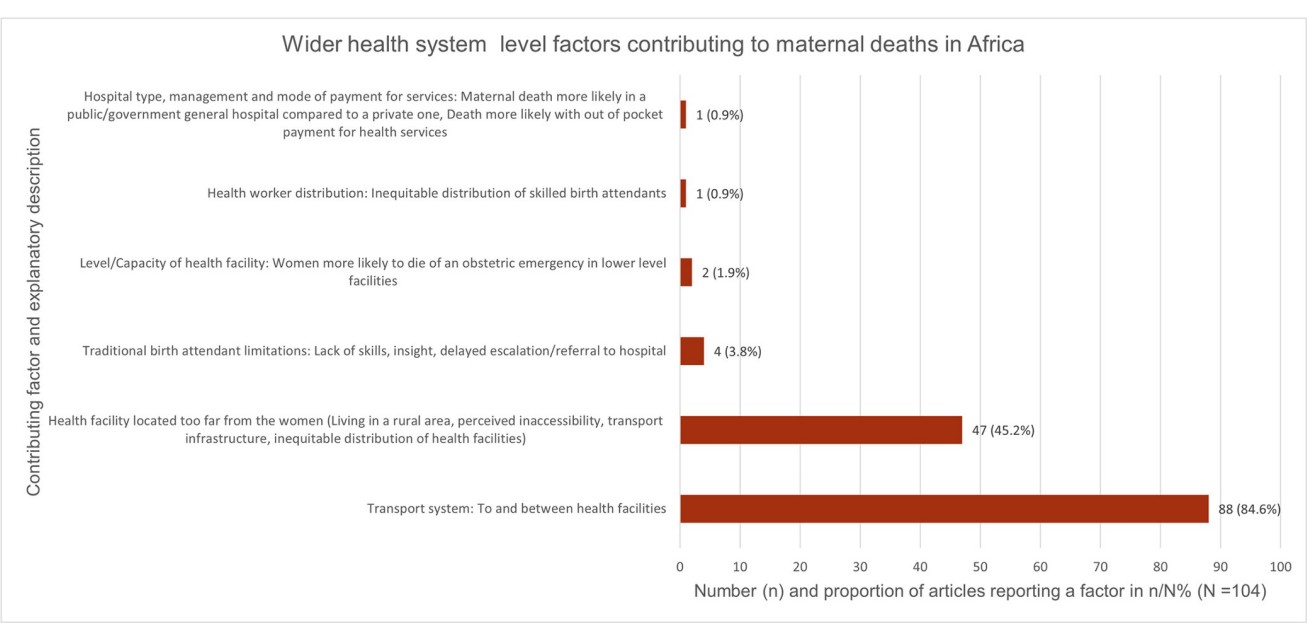

**Fig 5. Wider health system-level factors contributing to maternal deaths in Africa.** They are ranked from the most frequently reported (transport to and between health facilities) to the least frequently reported (hospital type, management, and mode of payment for health services). The number and proportion n (%) of articles reporting a specific factor is on the X-axis. The total number of included articles (N = 104) is the denominator. The factors and their explanatory descriptions are on the Y-axis.

gap in information as half of the African countries were not represented in our analysis due to a lack of eligible articles. This information gap is an important finding because where data are lacking, it would be difficult to formulate specific evidence-based strategic interventions for improving maternal health outcomes and tackling preventable maternal deaths. We could not analyse the strength of association for each contributing factor and maternal deaths as these data were rarely available within most primary research articles.

## Interpretation of findings

We set off to identify the factors contributing to maternal deaths in Africa and assess the frequency of their reporting in published literature using scoping review methods. While interpreting our findings, it is important to consider that we included articles with various study designs, including quantitative, qualitative, and mixed methods studies. Some of the contributing factors from the qualitative and mixed methods studies were based on the participants' perceptions. As such, perceptions are prone to perception bias [147]. Ideally, we should have extracted only the statistically significant contributory factors as they are accurate. However, doing so would have limited our scope of mapping all published contribution factors using primary studies.

By analysing the factors contributing to maternal deaths in Africa from primary studies, we have provided further insight into the persistent problem of maternal deaths in Africa and presented the contributing factors in an actionable format to better inform the design of intervention strategies and allocating resources for tackling preventable maternal deaths. Overall, our findings complement the findings of existing studies. Our review found contributing factors across all the three levels where national and regional governments across Africa could play a role in reducing maternal deaths. For example, investing in an efficient and reliable transport network could translate into improved access to health facilities by women during and after

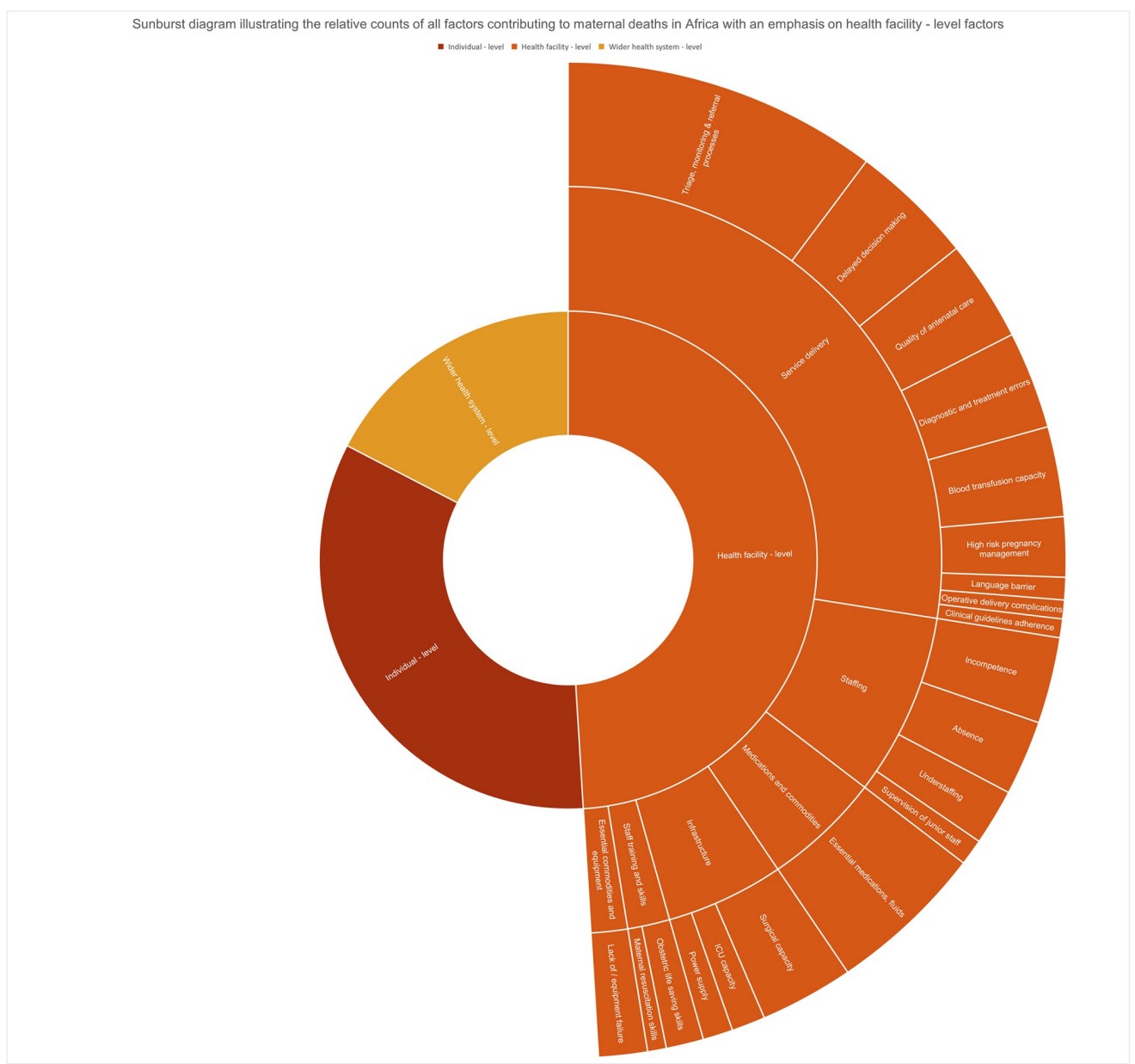

**Fig 6. A comparison of the various factors contributing to maternal deaths in Africa, emphasising health facility-level factors.** The total counts were 822 (Individual level 276 (33.6%), health facility level 403 (49.0%), and wider health system level 143 (17.4%). The health facility-level factors were service delivery 226 (56.1%), staffing 65 (16.1%), infrastructure 42 (10.4%), medications 40 (9.9%), staff training and skills 15 (3.7%) and essential commodities and equipment 15 (3.7%). See S2 Table for further details.

pregnancy and childbirth and prevent maternal deaths. Similarly, investing in gender equity and empowerment initiatives may help resolve some individual and health facility-level contributing factors [24]. Social scientists using political economy methods explored the explanations for the variations in maternal mortality rates between countries. They established that country-level political commitment differences might explain these MMR variations between developing countries [11]. Political goodwill and commitment are vital for the success of health initiatives.

Although we presented these contributing factors under three levels, they are interrelated. Other investigators reported a significant relationship between MMR and socio-economic, health care, and morbidity indicators and an inverse and significant correlation of the MMR with better antenatal care coverage, skilled birth attendants, improved water source and sanitation, higher adult literacy rate and Gross National Income per capita [19,20]. Systematic review evidence found that the level of skilled birth attendance and type of hospital accounted for 44% of the variation in facility-level MMRs [25]. Country per capita expenditure on health, the density of healthcare staff per 10,000 population, lower percentage of women with skilled birth attendance, access to an improved water source, fertility rate, prevalence of human immuno-deficiency virus (HIV) infections and Acquired Immunodeficiency Syndrome (AIDS), and population density may also explain MMR variations. [25] Although this systematic review was published in 2014, its findings are similar to the wider health system–level contributing factors we identified in our review. Any initiatives to improve maternal health outcomes and tackle preventable maternal deaths should consider this interrelation of contributing factors.

Our study found that health facility–level contributing factors accounted for nearly half of all factors contributing to maternal deaths. Our finding is similar to a statistical modelling population-level study that reported that medical/health determinants have the greatest impact on maternal mortality in sub-Saharan Africa, followed by socio-economic determinants [148]. Strategies that target addressing health facility–level factors have the potential to positively impact the efforts aimed at improving maternal health outcomes and tackling preventable maternal deaths in Africa.

Other factors that may be important to address in many African countries include the problem of armed conflict. Armed conflict is a wider health system-level contributing factor to maternal deaths. Conflict displaces women and their families from their homes and disrupts the healthcare delivery systems, maternal death review, reporting, and research activities. Therefore, the true burden of maternal deaths in areas with armed conflict may be highly underestimated or unreported. Some countries where we did not identify eligible articles have been in episodes of armed conflict. Armed conflicts disrupt maternal health services, with countries facing conflict having a higher risk of maternal deaths [21].

Another important wider health system factor to consider is the HIV-AIDs pandemic which disproportionately affected countries in sub-Saharan Africa more than other regions of the world, leading to direct and indirect increases in maternal deaths [149]. The negative effects of the HIV/AIDs pandemic on families and health systems in African countries persist to date.

Finally, regarding the individual level contributing factors, majority of the factors arise not due to the fault of the women but due to failures by the state to cater for the essential needs of its citizens, promotion of human rights, gender equity and empowerment, an enabling environment and functional healthcare systems. The obstetric transition concept illustrates five stages which may explain the differences in the magnitude and proportions (direct vs indirect) of maternal deaths seen between countries and why countries must promote social development, tackle inequity, support health system strengthening and improvements in quality of care, so as to improve maternal health outcomes and tackling of preventable maternal deaths [8].

## Strengths and limitations

The main strength of our scoping review is the detailed search strategy. Our search strategy sought articles from Africa-specific databases such as African Index Medicus, and African Journals Online (AJOL). It included articles in French—a common language in multiple

countries in West Africa. An additional strength is a mitigation against article selection and processing bias by following the Joanna Briggs Institute (JBI) updated guidance for conducting scoping reviews and the PRISMA Extension for Scoping Reviews (PRISMA-ScR) checklist and explanation [29,150].

One limitation emanates from the wide variety of study designs of the included articles. Some mixed methods and qualitative studies reported perceived contributor factors which may differ from real-world contributory factors. We also did not report the statistical significance of each contributory factor as these data were not universally available.

Another limitation emanates from the fact that our study sought to identify the contributing factors from published literature which is a potential for publication bias. There could be other factors that are either unexplored or reported in grey literature.

Finally, we recognise that we extracted the contributing factors from published articles and depended on the author's reporting of the determinants. The fact that we did not examine the primary data for each of the included articles may mean that any errors may be carried forward to our own review. This limitation exposes our work to reporting bias. We were also not privy to the various contexts where the primary studies were conducted and author limitations risking passing on bias arising from positionality. Despite these limitations, our scoping review has identified important factors contributing to maternal deaths in Africa, which is an important step for any future research assessing the strengths of association of each one.

## Implications for clinicians, policymakers, and governments in African countries

Although we synthesised the factors contributing to maternal deaths into individual, health-facility, and wider health system levels, we acknowledge that the three levels are not mutually exclusive. The contributing factors are interrelated, an important consideration while designing interventions to improve maternal healthcare delivery and tackle preventable maternal deaths. Examples of interventions that can positively impact all the three levels of contributory factors we conceptualised include a government-led initiative to improve the transport network and infrastructure and gender equity and empowerment initiatives.

Some positive changes may be realised by examining the health facility level processes that could improve efficiency, avoid wastage of resources, improve maternal health outcomes, and tackle preventable maternal deaths.

In summary, clinicians, policymakers, governments, patients, families, and the general public can make positive contributions toward tackling preventable maternal deaths. Such an effort requires deliberate investment in comprehensive and coordinated strategies. These recommendations are in synchrony with the multi-agency strategies to end preventable maternal mortality (EPMM) [18].

## Suggestions for future research

From our research, we identified information gaps from which we make these recommendations:

Future research should aim to establish the factors that contribute to maternal deaths in countries from where no articles were included, especially those affected by armed conflict.

Future studies should examine the relationship between the contributory factors and maternal deaths to identify the statistically significant ones.

Future studies should examine other possible explanations for the variations found in the contributing factors and maternal mortality ratios between countries and at subnational levels such as provinces, districts, and health facilities. This approach may identify best performers (positive outliers) whose practices could be adopted by neighbouring poor performers within the same setting and lead to an overall improvement in maternal health outcomes (positive deviance approach) [151]. This approach, by extension, would reduce preventable maternal deaths.

## Conclusion

We identified, synthesised, and classified multiple factors contributing to maternal deaths in Africa into three levels: Individual, health facility and the wider health system levels. The health facility-level of contributory factors were the most reported.

We also proposed the most important contributing factors s that clinicians, policymakers, and governments need to address to achieve better and quicker results in tackling the problem of preventable maternal deaths in African countries.

To inform future research, we highlighted areas where information gaps remain.

## Supporting information

**S1 Checklist.**
(PDF)

**S1 Table. Search strategy and outputs.**
(DOCX)

**S2 Table. Factors contributing to maternal deaths in Africa.**
(DOCX)

**S1 Data. Included studies and contributory factors.**
(XLSX)

**S2 Data. Articles excluded at the full-text review stage.**
(XLSX)

## Acknowledgments

Library and Knowledge Service, Birmingham Women's and Children's NHS Foundation Trust and University Librarians, University of Birmingham.

## Author Contributions

**Conceptualization:** Francis G. Muriithi, Ioannis D. Gallos.

**Data curation:** Francis G. Muriithi, Ruth Gakuo, Kia Pope.

**Formal analysis:** Francis G. Muriithi, Aduragbemi Banke-Thomas, Ioannis D. Gallos.

**Funding acquisition:** Arri Coomarasamy, Ioannis D. Gallos.

**Investigation:** Francis G. Muriithi.

**Methodology:** Francis G. Muriithi, Aduragbemi Banke-Thomas.

**Project administration:** Francis G. Muriithi.

**Supervision:** Arri Coomarasamy, Ioannis D. Gallos.

**Writing – original draft:** Francis G. Muriithi.

**Writing – review & editing:** Francis G. Muriithi, Aduragbemi Banke-Thomas, Ruth Gakuo, Kia Pope, Arri Coomarasamy, Ioannis D. Gallos.

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
