## [Decision Letter · Decision Letter 0]

21 Apr 2022

PGPH-D-22-00285

Individual, health facility and wider health system determinants of preventable maternal deaths in Africa: A scoping review

Dear Dr. Muriithi,

Thank you for submitting your manuscript to PLOS Global Public Health. After careful consideration, we feel that it has merit but does not fully meet PLOS Global Public Health’s publication criteria as it currently stands. Therefore, we invite you to submit a revised version of the manuscript that addresses the points raised during the review process.

In revising the manuscript, please pay particular attention to reviewer comments on clarity of the research objective, clarity of the levels of determinants and individual determinants identified, and the process for extracting determinants. Be sure that all conclusions are based on a cautious interpretation of the findings in the context of not just what has been studied but what has not been studied.

Please submit your revised manuscript by . If you will need more time than this to complete your revisions, please reply to this message or contact the journal office at globalpubhealth@plos.org. Please include the following items when submitting your revised manuscript:

We look forward to receiving your revised manuscript.

Kind regards,

Hannah Hogan Leslie, PhD

Academic Editor

Journal Requirements:

1. Your co-authors:

Aduragbemi Banke-Thomas -a.bankethomas@gre.ac.uk

Ruth Gakuo -ethagakuo@gmail.com

Kia Pope -kiapope@yahoo.com

Ioannis D Gallos -I.D.Gallos@bham.ac.uk

,have not confirmed authorship of the manuscript. We have resent them the authorship confirmation email; however please check that the above email address for them is correct and follow up personally to ensure they confirm. 

Please note that we cannot proceed your manuscript  until we have received confirmations from all co-authors.

2. Please update the completed 'Competing Interests' statement, including any COIs declared by your co-authors. If you have no competing interests to declare, please state "The authors have declared that no competing interests exist". 

3. Please amend your detailed Financial Disclosure statement. This is published with the article, therefore should be completed in full sentences and contain the exact wording you wish to be published.

ii). State the initials, alongside each funding source, of each author to receive each grant.

4. In the online submission form, you indicated that "All the data relating to this manuscript is fully available and without restriction as supplementary files. All articles (included and excluded) are archived on Covidence and are available on request to the corresponding author.". All PLOS journals now require all data underlying the findings described in their manuscript to be freely available to other researchers, either 1. In a public repository, 2. Within the manuscript itself, or 3. Uploaded as supplementary information.

5. Please provide us with a direct link to the base layer of the map used in Figures 2 & 6 and ensure this location is also included in the figure legend. 

Please note that, because all PLOS articles are published under a CC BY license (creativecommons.org/licenses/by/4.0/), we cannot publish proprietary maps such as Google Maps, Mapquest or other copyrighted maps. If your map was obtained from a copyrighted source please amend the figure so that the base map used is from an openly available source.

Please note that only the following CC BY licences are compatible with PLOS licence: CC BY 4.0, CC BY 2.0  and CC BY 3.0, meanwhile such licences as CC BY-ND 3.0 and others are not compatible due to additional restrictions. If you are unsure whether you can use a map or not, please do reach out and we will be able to help you. 

The following websites are good examples of where you can source open access or public domain maps:

6. We have noticed that you have uploaded supporting information but you have not included a list of legends.  Please add a full list of legends for all supporting information files (including figures, table and data files) after the references list. 

Reviewers' comments:

Reviewer's Responses to Questions

**Comments to the Author**

1. Does this manuscript meet PLOS Global Public Health’s publication criteria? Is the manuscript technically sound, and do the data support the conclusions? The manuscript must describe methodologically and ethically rigorous research with conclusions that are appropriately drawn based on the data presented.

Reviewer #1: Yes

Reviewer #2: Yes

Reviewer #3: Yes

Reviewer #4: Partly

2. Has the statistical analysis been performed appropriately and rigorously?

Reviewer #1: Yes

Reviewer #2: Yes

Reviewer #3: N/A

Reviewer #4: N/A

3. Have the authors made all data underlying the findings in their manuscript fully available (please refer to the Data Availability Statement at the start of the manuscript PDF file)?

Reviewer #1: Yes

Reviewer #2: Yes

Reviewer #3: Yes

Reviewer #4: Yes

4. Is the manuscript presented in an intelligible fashion and written in standard English?

Reviewer #1: Yes

Reviewer #2: Yes

Reviewer #3: Yes

Reviewer #4: Yes

5. Review Comments to the Author

Reviewer #1: Thank you for the opportunity to review this important and rigorously conducted study.

I have some minor suggestion and one major point for the authors to consider.

Abstract

- Methods: mention the conceptual framework you used to summarise results

- Results: 24 out of 106 is not “Most of the articles (24) were from Nigeria”. Do you mean “The most commonly represented country was Nigeria”?

Main text

Introduction

- Line 88: “The reasons for the variation in MMR at national and sub-national levels remain largely unexhausted [11].” Is unexhausted the right verb here?

- Line 115: “Studies that utilise data sources based on statistically modelled MMR estimates are not always accurate compared to estimates 117 derived from primary studies [27]” I think you mean “Studies of determinants of maternal mortality which use data sources…” to be precise. Otherwise it could be confused with studies of causes of maternal mortality.

Methods

- Line 148: “We excluded inaccessible articles if no full-text versions were received from the corresponding authors and the University of Birmingham Library service.” I think here you mean “or” the library; you don’t need the full text from both sources (and).

- Line 155: “WHO 156 definition of determinants of health …” This would be a nice point at which to show a visual – the WHO framework and listing and your 3 combined categories?

Results

- Figure 1 – Flowchart: this is all clear, except the box at the very end where 280 articles were excluded after full-text was reviewed – some of the reasons seem like they should have been identified in much earlier stages of the screening (not published between 1987-2021; ineligible context). What does “ineligible population” mean? You did not present any population-based inclusion/exclusion criteria in Methods. The same is for “ineligible concept” – what does that mean?

- Now that I looked at SM1 I think I know what you mean by population, context and concept. But since this is only in SM, maybe a good idea to include these words in line 135 to guide the reader in understanding Figure 1?

- Figure 2 – can you check that you include a comprehensive legend (including the colour grey) and that the names of all the countries where papers were found + the number of papers from each appear in full and are legible?

- Table 1 – do you mean “Stages of pregnancy” or “Time of death”?

- In the main text, you refer to “supplementary file”, but there are 3 supplementary files. Can you enusre to refer to the correct file, to make this process easy for readers?

- Line 188: Can you also add how many countries in Africa had 0 studies?

- THIS IS MY ONLY MAJOR POINT: In terms of Figure 3- I am not sure I understood clearly the determinants you are listing. So each of the included studies presented maternal deaths and their determinants, that I get. But how did you extract those determinants? Did you only extract those which were found significant, or all determinants which were studied, or any which were mentioned? I imagine the 106 studies ranged hugely in terms of sample size of the maternal deaths investigated, right? Can you provide an overview of the sample size distribution across the 106 studies? Did any of the studies also look at determinants by cause of maternal death? To understand the specific links between determinants and causes more clearly?

- What were the various methods through which the 106 studies investigated determinants (review of medical records? Logistic regression? Etc)? Not sure all this is clear from Table 1. This is important.

- Figure 5 – a lot of these factors seem to only be captured by 1 study. Is there some kind of ecological study which is investigating this (GDP, literacy in population)? I cannot see ecological study design in Table 1.

- Figures 3,4,5 – As a reader I would have prefered you to list the number of studies rather than the %. But you could easily also do both.

- Figure 6. Not sure the title is clear. What does “Pattern of level of determinant…” mean? What do you mean by pattern? The most commonly reported type of determinant? But were all 3 levels of determinants investigated by each study in each country? If the study only focused on one type of determinant or only had sample to investigate one, then they could only identify one of the 3… I think this is a large limitation, as we are not sure whether all three levels were even considered in each study and therefor had an opportunity to be found as determinants.

- Figure 6. The colours used for Health facility and individual level are very similar – can you consider a contrasting colour to differentiate them? Also, not all country names are legible in this map. Mozambique has no name at all.

Discussion

- Line 232: “Our review found that most of the individual-level determinants fit in the broad theme of gender equity and empowerment” Actually I disagree, based on Figure 3, I would say that most of what you call individual-level determinants (= woman/family blaming) are health systems factors. Take the first one – is it the fault of the woman that she lives far from a facility, or the fault of the health system that there are no accessible facilities nearby/no reliable transport options? Why should the women be blamed for where they live? You can say the same about delay in care-seeking, late ANC booking, SES and cost of healthcare, TBA use, perceived low quality of care, self-medication…. Sure, some of the factors are within the gender equity and empowerment as you point out, but many are intimately related to a dysfunctional health system. I am not proposing that you re-do the analysis – I am just suggesting that perhaps this warrants a serious mention in the Discussion. I found it fascinating (by which I mean enraging) just how many papers you found explore what the women/families did wrong, versus how the health system is failing them. We should call these research practices out and try to do better. Being high parity or being poor is not “good” reason to die. A functioning health system should be able to identify and help these women.

Reviewer #2: This is a relevant systematic review on determinants of maternal mortality in African countries. The title may be not aligned with the objective of the research. The authors have searched for articles on determinants of maternal mortality, but NOT on "preventable" maternal mortality. For this reason, the title should be changed. The classification of the determinants into "individual, health facility and the wider health system levels" is interesting and may be useful for policy makers and health care service managers to prevent maternal mortality in such context.

Reviewer #3: The research is relevant and desirable. It tries to address the problem of maternal mortality in Africa from a wider perspective, beyond the health facility.

The authors have successfully presented this factors as individual level, health facility level and wider health system level. Health facility level factors were identified as the commonest when inter country comparisons were made followed by health system levels.

I think that the way the manuscript was presented has made these identified factors appear to be mutually exclusive rather than closely inter-linked factors and this may erroneously affect wider policy or decision making for health care. For example, closeness to a health facility (individual level factor) may actually mean closeness to a desired hospital to access care due to the poor condition of the closest geographical health facility arising from poor allocation of resources or weak supervisory oversight of the health facilities (health system factor). The authors should acknowledge the inter relatedness of these factors when considering interventions to improve maternal health care delivery

In addition, they should highlight the lack of access to the primary data in this work as an important limitation because they may not be privy to the context in which some of the original data in the selected publications were collected. This may affect how they interpreted it.

Reviewer #4: Dear Dr. Leslie,

Thank you for the opportunity to review this interesting manuscript, in which the authors conduct a wide scoping review to identify reported determinants of maternal mortality in Africa. This study is strengthened by a rigorous methodology, including a well-defined search strategy and thorough extraction process. I think the findings of this review would be a useful contribution. However, the paper would be strengthened by a more accurate representation of the paper’s contribution to the literature, a more robust synthesis of the results in the main text, and a more thorough, clearer summary of evidence in the discussion. Please note that the flow chart figure did not come through clearly so I could not include it in my review.

Major comments

The authors state their objective with this scoping review is to “identify the most common determinants of maternal deaths in Africa and explore their relationship with the country level MMR.” However, this paper does not tackle the second part of that objective. Rather, it identifies the universe of determinants reported in the literature and assesses their frequency (still a valuable contribution). I suggest the authors better specify the objective and ultimate contribution this review can make.

I suggest the authors take great care in expressing the policy implications of this work. In multiple places (eg, line 111 of background) the authors suggest that they can attribute maternal deaths by cause and aid prioritization of maternal health interventions. However, identified causes are not necessarily the most prevalent or most salient causes; they are simply those most widely studied and reported in the literature. For example, 82% of health system determinants were related to transport; to me, this does not mean that transport is the highest priority for health system improvement. In the discussion, the authors state “The greatest proportion of effort and resources should be focused on addressing the most frequently reported determinants.” I do not think this research supports that conclusion. I recommend the authors be clear that it reflects only the available literature and more clearly specify how this review can inform future work.

In describing the existing literature and the niche this review fills, the authors describe statistically modeled data from multiple organizations vs primary data. This is not a clear description. Are the authors referring to other systematic reviews, scoping reviews, and meta-analyses? Better engagement of the available literature and how this scoping review contributes is warranted. Please also clarify what you mean by “statistically modeled data.”

It would be helpful for the authors to thoroughly describe the inclusion/exclusion criteria used for studies, in particular those used to determine whether a study included causes of maternal mortality and which it reported on. Were criteria derived from the mentioned WHO definition?

The authors do a nice job describing the sources of evidence but need to build out the synthesis of results, rather than reporting only the number of identified studies and specifying further in figures alone. It would help the reader and add important context if authors were to specify in writing key determinants identified at each level. Similarly, the authors should describe the categories of determinants they identified and mention in the figures (eg, “suboptimal service delivery”) and explain how they define and categorize these.

The discussion makes some useful points but the flow and organization are somewhat difficult to follow. A more robust discussion of the main themes and relevance to key stakeholders is required. Please consider reorganizing and taking a deeper dive into the concepts and themes in the identified studies.

Minor comments

Abstract

I suggest clarifying the sentence beginning line 33—it sounds like the authors are claiming that existing studies of determinants of maternal mortality only use certain data from the listed organizations, which isn’t the case.

Background

Line 85: Please clarify what you mean by “interventions, omissions.”

Line 87: Causes are “largely similar” to what? Between countries?

Line 89: Please clarify what you mean by “unexhausted.” Do you mean the literature does not adequately capture all possible determinants of mortality?

I suggest cutting down paragraph 3 of the background (beginning line 90).

Methods

I recommend including more description of the search strategy, or at minimum some example search terms in the main text, even if full search hedges are in supplementary materials.

Please justify exclusion of systematic reviews—previous reviews can be valuable sources.

Please describe what data was extracted from each of the studies.

Discussion

The limitations section does not currently address any limitations.

Figures

Within each bar chart presenting identified determinants, I recommend ordering by type of determinant and then by magnitude.

6. PLOS authors have the option to publish the peer review history of their article (what does this mean?). If published, this will include your full peer review and any attached files.

**Do you want your identity to be public for this peer review?** For information about this choice, including consent withdrawal, please see our Privacy Policy.

Reviewer #1: No

Reviewer #2: No

Reviewer #3: **Yes: **Dr Michael Chudi Ezeanochie

Reviewer #4: No

---

## [Decision Letter · Decision Letter 1]

15 Jun 2022

PGPH-D-22-00285R1

Individual, health facility and wider health system factors contributing to maternal deaths in Africa: A scoping review

Dear Dr. Muriithi,

Thank you for submitting your manuscript to PLOS Global Public Health. After careful consideration, we feel that it has merit but does not fully meet PLOS Global Public Health’s publication criteria as it currently stands. Therefore, we invite you to submit a revised version of the manuscript that addresses the points raised during the review process. We would like you to revise the conclusions in the abstract to comment on the most frequently reported factors and the data gaps, rather than the relationships between factors which is an ancillary point in the discussion. No other revisions are requested; the reviewers and I appreciated your attention to the points raised on initial review.

Please submit your revised manuscript by . If you will need more time than this to complete your revisions, please reply to this message or contact the journal office at globalpubhealth@plos.org. Please include the following items when submitting your revised manuscript:

We look forward to receiving your revised manuscript.

Kind regards,

Hannah Hogan Leslie, PhD

Academic Editor

Journal Requirements:

Additional Editor Comments (if provided):

Please consider revising the conclusions in the abstract to better match the main findings of the paper.

Reviewers' comments:

Reviewer's Responses to Questions

**Comments to the Author**

1. If the authors have adequately addressed your comments raised in a previous round of review and you feel that this manuscript is now acceptable for publication, you may indicate that here to bypass the “Comments to the Author” section, enter your conflict of interest statement in the “Confidential to Editor” section, and submit your "Accept" recommendation.

Reviewer #1: All comments have been addressed

Reviewer #4: All comments have been addressed

2. Does this manuscript meet PLOS Global Public Health’s publication criteria? Is the manuscript technically sound, and do the data support the conclusions? The manuscript must describe methodologically and ethically rigorous research with conclusions that are appropriately drawn based on the data presented.

Reviewer #1: Yes

Reviewer #4: Yes

3. Has the statistical analysis been performed appropriately and rigorously?

Reviewer #1: Yes

Reviewer #4: N/A

4. Have the authors made all data underlying the findings in their manuscript fully available (please refer to the Data Availability Statement at the start of the manuscript PDF file)?

Reviewer #1: Yes

Reviewer #4: Yes

5. Is the manuscript presented in an intelligible fashion and written in standard English?

Reviewer #1: Yes

Reviewer #4: Yes

6. Review Comments to the Author

Reviewer #1: (No Response)

Reviewer #4: The authors have adequately addressed my comments on the original manuscript.

7. PLOS authors have the option to publish the peer review history of their article (what does this mean?). If published, this will include your full peer review and any attached files.

**Do you want your identity to be public for this peer review?** For information about this choice, including consent withdrawal, please see our Privacy Policy.

Reviewer #1: No

Reviewer #4: No

---

## [Editor Report · Decision Letter 2]

24 Jun 2022

Individual, health facility and wider health system factors contributing to maternal deaths in Africa: A scoping review

PGPH-D-22-00285R2

Dear Dr Muriithi,

We are pleased to inform you that your manuscript 'Individual, health facility and wider health system factors contributing to maternal deaths in Africa: A scoping review' has been provisionally accepted for publication in PLOS Global Public Health.

Best regards,

Hannah Hogan Leslie, PhD

Academic Editor